# An Experimental Model of Neuro–Immune Interactions in the Eye: Corneal Sensory Nerves and Resident Dendritic Cells

**DOI:** 10.3390/ijms23062997

**Published:** 2022-03-10

**Authors:** Laura Frutos-Rincón, José Antonio Gómez-Sánchez, Almudena Íñigo-Portugués, M. Carmen Acosta, Juana Gallar

**Affiliations:** 1Instituto de Neurociencias, Universidad Miguel Hernández—Consejo Superior de Investigaciones Científicas, 03550 San Juan de Alicante, Spain; l.frutos@umh.es (L.F.-R.); ainigo@umh.es (A.Í.-P.); mcarmen.acosta@umh.es (M.C.A.); juana.gallar@umh.es (J.G.); 2The European University of Brain and Technology-NeurotechEU, 03550 San Juan de Alicante, Spain; 3Instituto de Investigación Biomédica y Sanitaria de Alicante, 03010 Alicante, Spain

**Keywords:** corneal nerves, dendritic cells, neuro-immune interactions, ocular inflammation, ocular pain, animal models

## Abstract

The cornea is an avascular connective tissue that is crucial, not only as the primary barrier of the eye but also as a proper transparent refractive structure. Corneal transparency is necessary for vision and is the result of several factors, including its highly organized structure, the physiology of its few cellular components, the lack of myelinated nerves (although it is extremely innervated), the tightly controlled hydration state, and the absence of blood and lymphatic vessels in healthy conditions, among others. The avascular, immune-privileged tissue of the cornea is an ideal model to study the interactions between its well-characterized and dense sensory nerves (easily accessible for both focal electrophysiological recording and morphological studies) and the low number of resident immune cell types, distinguished from those cells migrating from blood vessels. This paper presents an overview of the corneal structure and innervation, the resident dendritic cell (DC) subpopulations present in the cornea, their distribution in relation to corneal nerves, and their role in ocular inflammatory diseases. A mouse model in which sensory axons are constitutively labeled with tdTomato and DCs with green fluorescent protein (GFP) allows further analysis of the neuro-immune crosstalk under inflammatory and steady-state conditions of the eye.

## 1. Introduction

Several studies have determined the influence of the nervous system on the immune system, but conversely, the influence of immune cells on the nervous system, besides their protective role, has not been so widely studied. The various functions of resident immune cells have been assessed in many ocular pathological conditions and diseases, however, their contribution to the maintenance of corneal tissue and sensitivity remains elusive. Corneal DC depletion during steady-state produces a reduction in the nerve ending density in the center of the cornea, as well as epithelial defects and delayed nerve regeneration [1]. Although previous results suggest that intraepithelial DCs and sensory nerves have intimate connections and are functionally interdependent, no functional or behavioral studies have been done so far to confirm this functional neuro–immune interaction.

## 2. The Cornea

The cornea and the conjunctiva constitute the eye tissues exposed to the environment. The cornea is an avascular connective tissue that is crucial, not only as the primary barrier for the eye but also as a powerful refractive structure. As a barrier, the cornea provides structural integrity to the ocular globe and protects its inner components from infectious agents and physical injury or chemical insults. On the other hand, as a refractive structure, it has two key properties: refractive power (for light refraction) and transparency (for light transmission) [2].

The shape of the human cornea is prolate, which creates an aspheric optical system [3]. The human cornea has a refractive index of 1.376 and a dioptric power higher than 40 dioptres, about 2/3 of the total ocular power [4]. It is 540–700 µm thick, being thinner in the center and thicker in the periphery [2]. The human cornea measures about 11 mm vertically and 12 mm horizontally [5] covering the anterior 1/6th of the ocular surface and is organized into 5 layers (Figure 1): epithelium, Bowman’s layer, stroma, Descemet’s membrane, and endothelium. Among species, the cornea keeps the same general structure, with differences in thickness and presence or absence of Bowman’s layer.

Transparency of all ocular structures is crucial for vision. Transparency of the cornea is the result of many factors including its highly organized anatomical structure, the physiology of its cellular components, the lack of myelination of nerves inside the cornea, its tightly controlled (de)hydration state and the absence of blood and lymphatic vessels, among others. Since vision plays a critical role in obtaining information from the external environment, the cornea also has specific characteristics that ensure its own protection against injury. One of the most important features in this regard is the high sensitivity of the cornea to external insults provided by its extremely rich sensory innervation.

Taking all the corneal characteristics into account, it is clear that the cornea is an ideal model for studying the interactions between corneal nerves and the few cell types present in this quite simple structure. Particularly, the cornea represents a perfect scenario to study neuro–immune interactions for different reasons: the cornea is densely innervated [6,7] with sensory nerves easily accessible for electrophysiological recordings whose functional properties have been described in detail [8,9]; the cornea is fully transparent [10,11], which means that fluorescence data could be gathered at high resolution for both in vivo and ex vivo experiments and finally, the cornea is avascular [12,13], which allows distinguishing the contribution of resident immune cells from that of immune cells migrating from blood vessels.

### 2.1. Corneal Structure

#### 2.1.1. Epithelium

The corneal epithelium composed of a single layer of basal cells and several stratified and non-keratinized cell layers (Figure 1, inset), constitutes the first protective ocular barrier against external threats. In humans, it is 50 µm thick, representing around 8% of the total thickness of the cornea [14].

The outermost corneal epithelium constitutes 2–3 layers of squamous cells [3] (Figure 1, inset). These cells are flat and polygonal and have apical microvilli which, in turn, are covered by a fine glycocalyx consisting of membrane-associated mucins, including MUC1, MUC4, and MUC16 [15]. These squamous cells maintain tight junctions with their neighbors, which is essential for their function as a barrier to prevent large molecules or microbes from entering the deeper corneal layers. Beneath the superficial layers of squamous cells are the mid-epithelial layers of wing cells (Figure 1, inset). Wing cells are less flattened but have tight lateral junctions with their neighbors very similar to those observed in the squamous cells [3]. Finally, basal cells constitute the deepest cell layer of the corneal epithelium (Figure 1, inset). This layer, around 20 µm thick, constitutes a single layer of columnar epithelial cells that are connected through gap junctions and desmosomes and are attached to the underlying basement membrane by hemidesmosomes, preventing the epithelium from separating from the other corneal layers [3].

In mice, the epithelium contributes approximately 30% to the total corneal thickness. The stratified layout of the murine corneal epithelium is consistent with the description of this epithelium in the mammalian cornea. However, the mouse corneal epithelium consists of approximately twice the layers of cells compared with the human epithelium, with a higher number of squamous cells [16].

Corneal epithelial cell layers turn over every 7–10 days [17] following a delicate balance between superficial cell shedding and cell proliferation and migration from basal cells. Basal progenitor epithelial cells from the limbus (limbal stem cells) migrate towards the center of the cornea where they differentiate into transient amplifying cells (TA). TA basal cells, which are two horizontal progenies from the stem cells, migrate from the limbus to the periphery of the cornea to reach the center and undergo mixed proliferation (one daughter cell is retained in the basal cell layer, and the other moves into the middle layers of the epithelium). Once TA cells reach the end of their proliferative capacity, they become basal epithelial cells. Basal epithelial cells undergo vertical proliferation in two daughter cells resulting in two wing cells and eventually into two squamous cells that will be later shed by blinking. This concept was coined the X, Y, Z hypothesis of Thoft and Friend [18] where X is the basal corneal epithelial cell horizontal migration and vertical terminal proliferation, Y is the limbal cell proliferation and migration and Z is the squamous corneal epithelial cell shedding.

Between the corneal epithelium and the next corneal layer, the stroma, there is the epithelial basement membrane synthesized by epithelium basal cells. This membrane, also called the basal lamina, is approximately 0.05 µm thick and comprises laminin, type IV collagen, heparan sulfate, and fibronectin. Basal lamina serves as a scaffold for epithelial cell movement and attachment, and it is composed of two distinct layers when observed by electron microscopy: the more external *Lamina lucida* and a thicker more internal *Lamina densa* [2].

#### 2.1.2. Bowman’s Layer

Bowman’s layer in the human cornea is an acellular condensate of collagen types IV, V, VI, and VII arranged randomly [4] that help the cornea to maintain its shape [3]. This layer is approximately 15 µm thick and is associated with the basal lamina of the epithelium and continues with the stroma. Different roles have been ascribed to this layer of the cornea that is not present in all species, such as that it provides some kind of barrier function against the passage of macromolecules, such as medium and large size proteins, or that it is responsible for a substantial portion of the biomechanical rigidity of the cornea. However, there are also studies that conclude the opposite and the exact function of this layer remains unclear (see Wilson 2020 for a review [19]). It has been hypothesized that Bowman’s layer develops because of cytokine-mediated interactions occurring between corneal epithelial cells and the underlying keratocytes, including negative chemotactic and apoptotic effects on the keratocytes by low levels of cytokines, such as interleukin-1α [20]. Bowman’s layer is highly resistant to damage, but it cannot regenerate after injury and may result in a scar [21].

It is worth mentioning that in mice, subepithelial collagen fibers are also arranged randomly forming what would be a thin Bowman’s layer. However, some authors believe that this is not a true layer [22] but an adaptation of the stromal tissue [16].

#### 2.1.3. Stroma

Corneal stroma comprises 90% of the total corneal thickness [2]. It is mainly composed of water (78%), an organized structural network of types I and V collagen fibers (80% of corneal stroma’s dry weight), keratocytes, and extracellular matrix.

The stromal collagen fibers are arranged in parallel bundles (fibrils) that are packed in parallel arranged layers (lamellae). In turn, each of these layers is arranged at right angles relative to fibers in adjacent lamellae, and this precise organization results in stromal transparency as it reduces forward light scatter [23].

Keratocytes, the major cell type of corneal stroma, are sandwiched between collagenous lamellae, mostly in the anterior stroma. These stellate-shaped cells are connected to each other through gap junctions present on their numerous dendritic processes [24]. Keratocytes are involved in maintaining the extracellular matrix environment and stromal composition as they are able to synthesize glycosaminoglycans, collagen molecules, and matrix metalloproteases (MMPs) [3]. As a first response to stromal injury, keratocytes are activated and migrate taking on a fibroblast-like appearance [25] and within 1–2 weeks of the initial insult, myofibroblasts enter the injured area and become involved in the stromal remodeling which can take months or even years to complete [3].

In mice, the corneal stroma accounts for two-thirds of the total corneal thickness. In these animals, stromal collagen fibers have a diameter of 29 ± 4 nm [26] with keratocytes arranged parallel to the collagen bundles.

In the posterior part of the human corneal stroma, there is a distinct region that constitutes the separation between the stroma and the Descemet’s membrane. This pre-Descemet or Dua’s layer is acellular and composed of 5 to 8 lamellae of predominantly type-1 collagen bundles arranged in transverse, longitudinal, and oblique directions [27].

#### 2.1.4. Descemet’s Membrane

Descemet’s membrane is primarily composed of types IV and VIII collagen fibrils as well as the glycoproteins fibronectin, laminin, and thrombospondin. It is less strong and stiff than the posterior stroma and is secreted by endothelial cells since the 8th gestation week [3]. Descemet’s membrane has a thickening rate of approximately 1 µm per decade of life: its thickness is around 4 µm at birth, while at the end of the normal lifespan Descemet’s membrane is around 10–15 µm thick [2]. In addition, this thickness can also increase focally or diffusely with injury (trauma or surgery) or disease, due to abnormal collagen deposition.

Descemet’s membrane in mice is more homogeneous and granular on the anterior chamber side, resembling a typical basal lamina, and it also becomes thicker with age [28].

#### 2.1.5. Endothelium

The corneal endothelium is a monolayer of hexagonal cells whose density and topography change continuously throughout life [3]. At birth, corneal endothelium is 10 µm thick and its cell density is approximately 3500 cells/mm^2^, however, this number decreases at approximately 0.6% per year [3]. Corneal endothelium maintains its continuity by migration and expansion of survival cells to cover the defect surface, so the percentage of hexagonal cells decreases (*pleomorphism*, that is, endothelial cells become different from each other in shape), while the coefficient of variation in cell area increases (*polymegathism*, that is, endothelial cells become different from each other in size, appearing large cells) [2].

The main function of the corneal endothelium is to maintain corneal transparency and health by regulating its hydration and nutrition [2]. Adjacent endothelial cells share extensive lateral interdigitations and possess tight and gap junctions along their apical and lateral borders, respectively, forming an incomplete barrier with a preference for the diffusion of small molecules [2]. The endothelium acts as a “leaky” barrier that allows passive fluid flow from the hypotonic corneal stroma to hypertonic aqueous humor through the osmotic gradient, maintaining the relatively dehydrated state of the stroma. Nevertheless, although this passive movement does not require energy, endothelial cells maintain the osmotic gradient by active transport of ions. In this context, the major transport protein found to be essential is Na^+^/K^+^-ATPase, present in the basolateral membranes of endothelial cells.

### 2.2. Corneal Innervation

The cornea is supplied by both sensory and autonomic nerves, being one of the most densely innervated tissues in the body [6]. The trigeminal nerve, the major sensory nerve of the head, has three different sensory branches: ophthalmic (V1), maxillary (V2), and mandibular (V3) [29,30]. Most nerves supplying the cornea are sensory nerves, and most of them have their origin in the ophthalmic branch of the trigeminal ganglion (TG). Furthermore, a little innervation from the maxillary branch has also been reported in the inferior cornea and the conjunctiva [31,32].

V1 branches into the frontal, the lachrymal, and the nasociliary nerves. In turn, the nasociliary nerve branches into two long ciliary nerves and a connecting branch with the ciliary ganglion, a parasympathetic ganglion that sends 5–10 short ciliary nerves carrying both trigeminal sensory nerve fibers from the nasociliary nerve and parasympathetic and sympathetic axons, the last from the superior cervical ganglion. The density of the sympathetic nerves varies significantly among different species [33], being higher in the cat and rabbit cornea [34,35,36] and very sparse in humans and other primates [34,37].

Short and long ciliary nerves enter the posterior globe medially and laterally to the optic nerve [30,38], penetrating the sclera. After that, they form a ring around the optic nerve and travel anteriorly in the suprachoroidal space towards the anterior segment of the eye [30], undergoing repetitive branching. When reaching the limbal area, some fibers innervate the ciliary body and the iris, while most fibers form a dense ring-like network that encircles the limbus around the cornea, giving rise to the limbal plexus [39]. The majority of nerve fibers in this plexus are believed to be vasomotor nerves innervating limbal blood vessels, while a variable number of nerve trunks enter the corneal stroma unrelated to blood vessels [40].

#### 2.2.1. Corneal Nerve Architecture

The architecture of corneal innervation has been studied for many years by a wide variety of methods, including light and electron microscopy, immunohistochemistry, and IVCM, among others. Moreover, it has been described among different species such as human [6,38], cat [36,41], guinea pig [42,43], and mouse [44,45]. Corneal innervation is anatomically organized into four levels from the penetrating stromal nerve trunks to the nerve terminals in the epithelium.

##### Stromal Nerves

Corneal stromal nerves enter the cornea radially through the corneoscleral limbus in the middle third of the stroma. In addition, other small nerve bundles enter the cornea more superficially in the episcleral and conjunctival planes, innervating the superficial stroma and the periphery of the corneal epithelium, respectively [39,41,42].

When entering the stroma at a depth of approximately 293 µm, myelinated axons (about 20%) lose their perineurium and myelin sheath [6] and run within the stroma as fascicles enclosed by a basal lamina and Remak Schwann cells (the non-myelinating Schwann cells) [30]. The distal branches of this arborization anastomose extensively form the anterior stromal nerve plexus, a complex network of nerve bundles and individual axons. The posterior half of the stroma and endothelium, on the contrary, are devoid of sensory nerve fibers [30].

In mice, stromal nerves do not enter the cornea radially at regular intervals as in humans. Stromal innervation is provided by nerve bundles entering into the cornea from four quadrants and branching irregularly to cover the entire cornea (Figure 2A,B).

In humans and higher mammals, the most superficial layer of the anterior stromal nerve plexus, immediately beneath Bowman’s layer, is referred to as the corneal subepithelial nerve plexus. The subepithelial plexus has a very high nerve density, but in general, it is denser in the peripheral and intermediate cornea, and less dense in the central cornea [38].

Anatomically, in the subepithelial plexus, there are two distinct types of nerve bundles [30,38]. One form a complex anastomotic meshwork of single axons and thin tortuous fascicles that do not penetrate Bowman’s membrane, while the second type consists of about 400–500 medium-sized, curvilinear bundles that turn 90° and penetrate Bowman’s layer and basal lamina mainly in the peripheral and intermediate cornea [6,38]. These nerve bundles terminate in bulb-like structures and divide into smaller ones in groups up to 20 subbasal nerve fibers, known as epithelial leashes [6,38]. These leashes, which are parallel to the ocular surface, anastomose extensively to form a dense subbasal nerve plexus (see below).

It is worth mentioning that while unmyelinated nerves maintain their Remak Schwann cell coating in the stroma, they shed them before penetrating the basal lamina. In this regard, it has been suggested that corneal epithelial cells function as surrogate Schwann cells for the subbasal and intraepithelial nerves in healthy conditions and after injury [46].

##### Subbasal Nerve Plexus

The corneal epithelium receives sensory nerve fibers either from the subepithelial plexus or directly from the conjunctival nerves [39,40]. The subbasal nerve plexus constitutes epithelial leashes from subepithelial nerves that anastomose extensively and interconnect repeatedly with one another such that they are no longer recognizable as individual leashes (Figure 2C,D), although leashes are less numerous and separated in the periphery [38]. The term “epithelial leash” was defined as a group of subbasal nerves that derive from the same parent anterior stromal nerve trunk [41,47,48], being a unique neuroanatomical structure only found in the cornea of most species, including humans.

The subbasal nerve plexus is a dense, homogenous nerve plexus situated between the basal epithelial cells and the basal lamina. In humans, it is formed by 5000–7000 nerve fascicles in an area of about 90 mm^2^ [49], with a total number of axons estimated to vary between 20,000 and 44,000 [6]. Morphologically, each leash consists of a variable number of straight nerve fibers, each containing 3–10 individual axons traveling up to several millimeters. Subbasal nerve fibers converge on a spiral whose center is called the vortex [50,51]. This vortex is also present in other species, including mice [43,44] and rats [52], and the mechanisms underlying its formation remain unclear. The most extended hypothesis is that nerves and basal epithelial cells advance in the same direction and velocity in a whorl-like pattern in response to chemotropic guidance, electromagnetic cues, and/or to population pressures [53,54].

##### Intraepithelial Nerve Terminals

Single nerve fibers arising from the subbasal plexus split off and turn 90° vertically as a profusion of terminal axons ascending between the epithelial cells, often with a modest amount of additional branching (Figure 2E–I) [30]. The term intraepithelial nerve terminal was defined by Carl Marfurt as the entire epithelial axon distal to its point of origin from a subbasal nerve and all its collateral branches and terminal expansions (the so-called nerve endings) [38]. Corneal epithelium innervation is extremely dense. It is estimated that the human central cornea contains around 5000–7000 nerve terminals per square millimeter, although this number changes throughout life and in ocular pathologies.

The intraepithelial nerve terminals innervate the corneal epithelium through all its layers. Those running between the basal and wing epithelium cells run in a horizontal direction and branch relatively infrequently [38], while intraepithelial nerve terminals that terminate within the more superficial cell layers are generally more complex (Figure 2E,G–I). Intraepithelial nerve terminals can be classified into three different types: simple, ramifying, and complex [29]. Simple terminals do not branch after leaving the subbasal plexus and end with a bulbar swelling within or below the superficial squamous cells [29,43] (Figure 2G). Simple terminals are more abundant in the central cornea. Ramifying terminals branch within the squamous cell layer into 3–4 branches that run horizontally for a hundred microns and that end in a single bulbar swelling like the simple terminals [29,43] (Figure 2H), being more numerous in the peripheral cornea [43]. Finally, axons forming the complex terminals start to branch within the wing cells layer and terminate with multiple larger bulbous endings within the wing and squamous cell layers (Figure 2I). Complex terminals are found in the central and the peripheral cornea, but their complexity and size are higher in the periphery [43].

Intraepithelial nerve terminals seem to be functionally different as immunocytochemical staining reveals differences in the expression of neuropeptides and neurotransmitters. In mice and guinea pigs, nerves that terminate in the basal epithelium and the outermost cell layers have simple endings immunopositive for Calcitonin Gene-Related Peptide (CGRP) and substance P (SP), suggesting that peptidergic simple nerve terminals correspond functionally to polymodal nociceptor nerve terminals [43]. On the other hand, complex nerve terminals are immunoreactive to TRPM8 (Transient receptor potential channel subfamily M member 8), supporting the idea that complex terminals correspond to cold thermoreceptor terminals [43]. More recent studies in guinea pig corneas also suggest that these intraepithelial nerve terminals can be distinguished morphologically as TRPM8-positive terminals are more complex than TRPV1-positive terminals [55].

**Figure 2 ijms-23-02997-f002:**
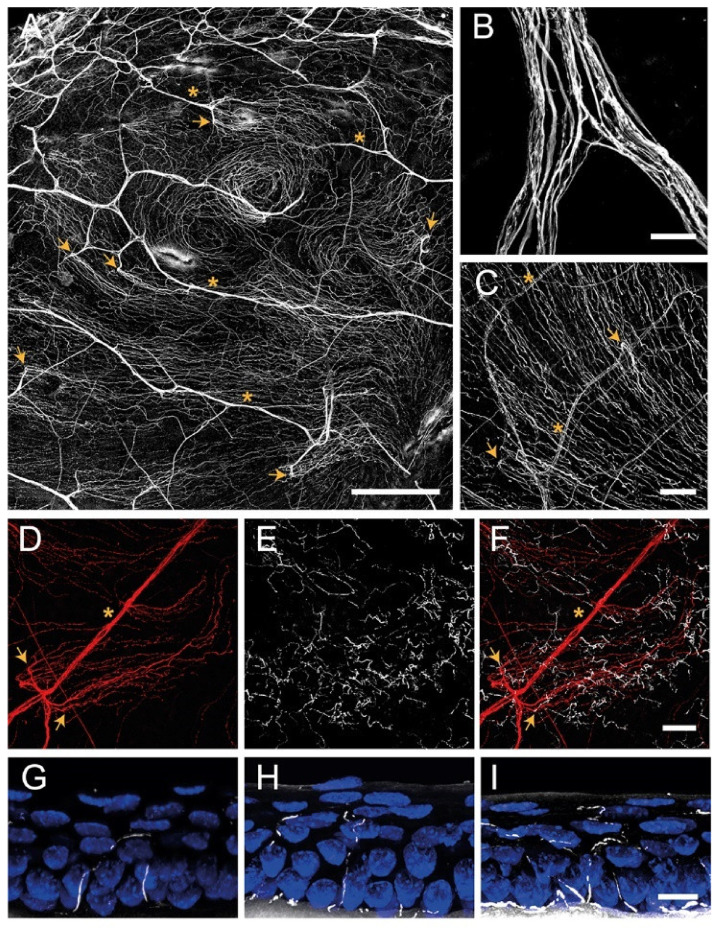
Confocal images of sensory nerves immunostained with anti–β tubulin III antibody in mouse cornea. (**A**) Sensory nerve trunks enter from the limbus into the stroma of the cornea where they ramify, giving rise to a dense subepithelial plexus (asterisks). (**B**) Detail of a stromal nerve trunk branching. (**C**,**D**) From the stroma, nerve fibers penetrate through the basal lamina (arrowheads) and form the subbasal plexus. Subbasal nerve fibers run parallel for a long distance within the epithelium basal cell layer. (**E**–**I**) Subbasal nerves give rise to terminal branches that ascend along their trajectory through the epithelial cells. According to the number of branches, three morphological types of corneal nerve terminals are identified: simple (**G**), ramified (**H**), and complex (**I**) nerve terminals. Scale bars: (**A**) 250 µm; (**B**) 25 µm; (**C**–**F**) 50 µm; (**G**–**I**) 10 µm. Methods: C57BL/6J eyes were fixed for 2 h at RT in methanol and DMSO (4:1), incubated 5 min at −20 °C in methanol, rehydrated, and washed in PBS. (**A**–**F**) Corneas were dissected, incubated 2 h at RT in blocking solution (5% goat normal serum, 1% BSA and 0.1%Triton X–100 in PBS) and 48 h at 4 °C with anti–β tubulin III antibody (1:500 in blocking solution; #801201, BioLegend, CA, San Diego, USA), rinsed and then incubated for 2 h at RT with AF555 (1:500 in PBS; #A32727, ThermoFisher Scientific, OR, Waltham, USA), washed in PBS and mounted with Fluoromount-G (ThermoFisher Scientific). (**G**–**I**) Eyes were cryoprotected (30% sucrose in PBS overnight at 4 °C), embedded in OCT, frozen in liquid nitrogen, cut on a cryostat in serial 15 mm thick sections, and mounted in slides. Tissue sections were rinsed in 0.03% Triton X–100 in PBS and then for 30 min in blocking solution followed by overnight incubation at 4 °C with anti–β tubulin III antibody in blocking solution, washing with PBS and incubation with AF555 in PBS for 2 h at RT. Afterward, slides were rinsed with PBS, incubated with Hoechst 33342 (10 μm/mL; #H1399, ThermoFisher Scientific), and coverslipped with Fluoromount–G. Images were collected using a laser scanning confocal microscope Zeiss LSM 880 (Oberkochen, DEU).

#### 2.2.2. Functional Types of Corneal Nerves

Electrophysiological recordings of sensory nerve fibers innervating the cornea have revealed the existence of different functional types of ocular sensory nerves, classically classified depending on the modality of stimulus by which they are activated [7,56]. Most corneal sensory nerves are the peripheral branches of medium or small trigeminal neurons with thin myelinated (A-delta) or unmyelinated (C) axons [56]. The external stimuli are transduced by their intraepithelial nerve terminals into a discharge of nerve impulses that encode the stimulus’ spatial and temporal characteristics. The impulse discharge is conducted by trigeminal neurons to the central nervous system, where sensory input is processed to finally evoke a sensation and is also used to regulate protective functions, such as tearing and blinking. Depending on the variable activation of the different classes of corneal sensory neurons, different sensations are evoked [7,57].

##### Mechanonociceptors

About 15% of corneal nerve fibers are mechanonociceptors, which express Piezo2 channels [58,59] and are activated exclusively by mechanical forces (Figure 3). Mechanonociceptors are usually A-delta fibers and produce a short-lasting impulse discharge in response to a sustained mechanical stimulus, therefore signaling the presence and velocity of change in the mechanical force, rather than its intensity or duration [60,61]. These relatively rapidly adapting mechanosensitive fibers contribute to the pain experienced when a foreign body touches the ocular surface [57].

##### Polymodal Nociceptors

The majority of corneal sensory fibers (around 70%) are polymodal nociceptors, which express a diversity of transducing ion channels in their nerve terminals, such as TRPA1, TRPV1, ASIC, and Piezo2, that allow them to be activated by noxious mechanical forces, heat (temperatures over 39 °C) and a wide variety of exogenous and endogenous chemicals (protons, ATP, prostaglandins, cytokines, etc.) [8,62] (Figure 3).

Polymodal nociceptors produce an irregular and repetitive discharge as long as the stimulus is maintained that is proportional to its intensity [8,63]. Moreover, under certain circumstances, polymodal nociceptors can be sensitized, developing an irregular low frequency long after the stimulus has disappeared [64]. In addition, sensitization produces a decrease in threshold and an increase in the firing frequency in response to a new stimulus [60,65]. Most corneal polymodal nociceptors are slow-conducting C-type fibers and are the origin of the ocular discomfort and pain sensations developed under pathological conditions, local inflammation, or injury [64,66].

##### Cold Thermoreceptors

The third class of corneal sensory fibers is cold thermoreceptors (10–15%), associated with A-delta and C nerve fibers. Cold thermoreceptors have spontaneous discharge and increase their firing rate in response to temperature reduction and osmolality increases [63,67,68] (Figure 3). Cold thermoreceptors are transiently silenced upon warming, although some of them restart firing in response to high temperatures (paradoxical response to heat) [7,69]. Cold thermoreceptors firing increases proportionally to the speed and magnitude of the corneal temperature reduction, as well as to the final static temperature [7]. When enough cold thermoreceptors are recruited with augmented tear evaporation, a conscious sensation of dryness is expected [56]. Cooling sensations with temperature reductions are increasingly unpleasant when higher temperature decreases are applied [57].

The activity of corneal cold thermoreceptors expressing TRPM8 is crucial in different mechanisms protecting the eye, such as lacrimation and blinking [7,70,71,72]. The deletion of TRPM8 channels produces both a decrease in basal tearing [71] and blinking [72] in mice, supporting the idea that sensory input of cold thermoreceptors is used by the CNS to regulate blinking and tearing. Aging induces changes in TRPM8 expression and activity, which correlates with the changes in tearing developed with age [73].

#### 2.2.3. Changes of Nerve Activity under Inflammation and after Injury

After inflammation or lesion, corneal sensory nerve activity is altered. Like in other tissues, corneal nociceptors (specially polymodal nociceptors) are sensitized [64,69,74,75,76,77,78,79], a functional state characterized by an increase in spontaneous activity, a reduction in the response threshold, and an increased response to stimulation. Sensitization constitutes the basis of spontaneous pain and hyperalgesia experienced during inflammation. Additionally, corneal nociceptors also contribute to the inflammatory processes of the ocular surface (a process known as neurogenic inflammation) [80,81] by releasing pro-inflammatory neuropeptides, such as SP and CGRP [6,82,83]. After injury, regenerating nociceptors present increased spontaneous activity due to the increased expression of specific types of Na^+^ channels by regenerating neurons [79].

Contrarily, cold thermoreceptors’ activity is decreased under inflammation [69,76] because the activity of TRPM8 channels is inhibited by inflammatory mediators, such as bradykinin through a G-protein [84]. During chronic tear deficiency, the activity of cold thermoreceptors is increased due to the increase in Na^+^ currents and the decrease in K^+^ currents [77].

### 2.3. The Cornea: An Immune-Privileged Tissue

The ocular surface is a mucosal surface in which the optical properties are critically important and where immune-mediated inflammation should not cause collateral damage [85]. Most of our current knowledge about the ocular surface immune-privileged arises from studies with corneal allografts, whose benefits have been widely described [86,87]. The cornea should perform an important balance between fighting infections and maintaining transparency in order to preserve vision. This immune-privileged tissue can be exposed to antigens, allergens, and pathogens without eliciting significant immune responses. The concept of ‘*immune privilege*’ was first coined by Medawar decades ago [88], however, it has been extensively reviewed since its initial conception [89,90]. Traditionally, the corneal immune privilege was ascribed to the lack of lymphatic and blood vessels and the lack of resident antigen-presenting cells (APCs) at basal conditions. Nevertheless, until today, different studies have demonstrated that the cornea is endowed with a significant number of resident APCs, such as macrophages and dendritic cells (DCs) [91,92,93,94,95]. In this regard, immune privilege means that even if immune cell activity is present, it is driven towards anti-inflammatory and tolerogenic immune responses [86].

Cells in the cornea, mostly epithelial cells, express and secrete different molecules that confer a tolerogenic profile upon DCs [85] or that promote regulatory activity or apoptosis in the T cells [86]. Among these molecules, the immunoregulatory factor Decay Accelerating Factor (DAF, also known as CD55) [96] or the apoptosis-inducing ligands FasL (FasLigand) and Programmed Death Ligand-1 (PDL-1) [85] seem to be crucial. In addition to suppressing T cell effector responses by producing their apoptosis, these ligands also allow regulatory T cells (Tregs) to function since Tregs are more resistant to FasL-induced apoptosis [97] and are activated by PDL-1 [98,99].

Along with the previously described mechanisms that contribute to the cornea’s immune privilege, there is also a neuroregulation of ocular surface immunity by corneal nerves [85]. Different neuropeptides released by corneal nerves are involved in this process. Vasoactive intestinal polypeptide (VIP) downregulates TLRs, inhibits chemokine expression, induces tolerogenic DCs, and limits the release of pro-inflammatory cytokines [100], while CGRP inhibits the production of inflammatory cytokines by macrophages and also the maturation of DCs [101].

## 3. Dendritic Cells

### 3.1. An Overview of Dendritic Cells

First discovered in mouse spleen in the 1970s [102], DCs are a heterogeneous group of professional APCs that induce naïve T cell activation and T effector differentiation [93,103], playing an important role between innate and adaptive immune responses. DCs include members of different lineages that can be found in two different functional states: mature and immature [93,103]. Immature DCs are characterized by a high antigen-capture ability due to their high endocytic capacity, but a low T cell-stimulatory capability due to their low surface expression of co-stimulatory molecules and chemokine receptors [104,105]. DC maturation, which is triggered by tissue homeostasis disturbances, leads to a decrease in their endocytic activity but an increase in the expression of major histocompatibility complex class II (MHC-II) and co-stimulatory molecules, such as CD40, CD80, and CD86 [105,106,107,108], becoming powerful T cell stimulators in secondary lymphoid organs [109,110].

Mature DCs can induce specific CD8+ and CD4+ T cell responses [111]. When interacting with CD4+ T cells, DCs can induce their differentiation into different T helper (Th) subsets, such as Th1, required for immunity against intracellular pathogens and cancer [103,111,112,113,114], Th2, essential for driving immune responses against parasitic infections [111,112,115,116], or Th17, important for neutralizing bacterial and fungal pathogens [117,118]. T cell differentiation in each subtype is a complex phenomenon that can be influenced by the cytokines in the DC tissue of origin [119] or their maturation state [107].

In addition to their role in inducing specific CD8+ and CD4+ T cells responses, DCs are also able to induce and maintain immune tolerance at basal conditions [103,120,121,122]. These “tolerogenic DCs” are immature DCs that express less co-stimulatory molecules, upregulate the expression of inhibitory molecules, and secrete anti-inflammatory cytokines [123,124], being essential to preventing responses against healthy tissues [109,120,125].

In both humans and mice, DCs are identified by their high expression of MHC-II and CD11c [103]. However, DCs express other molecules that allow their classification into different subtypes that differ from their phenotypic markers and genetic profile. Overall, these subtypes are types 1 and 2 conventional DCs (cDCs), plasmacytoid DCs (pDCs), Langerhans cells (LCs), and monocyte-derived DCs (MCs).

DCs arise from CD34+ hematopoietic stem cells (HSCs) that give rise to lymphoid (LPs) and myeloid precursors (MPs) (Figure 4). MPs differentiate into monocyte and DC precursors (MDPs), which, in turn, give rise to monocytes and the common DC precursors (CDPs). CDPs can differentiate into the preclassical DCs (pre-cDCs), which are the progenitors of the two major cDC subpopulations, cDC1 and cDC2, or into pDCs [103,126] (Figure 4), although recent studies suggest that mouse pDCs predominantly originate from a distinct progenitor from cDCs [111,127]. LPs can also give rise to pDCs, however, this ontogenic pathway is not completely elucidated (Figure 4). Once in the blood, pDCs and cDCs can migrate to lymphoid and nonlymphoid tissues.

LCs derive primarily from fetal liver monocytes that colonize the skin during embryogenesis [128] and maintain themselves by local proliferation in response to macrophage growth factors and IL-32 [109].

Finally, MCs are cells with DC-like features that can be generated by mouse monocytes during steady-state in skin and mucosal tissues [129,130] and during inflammation [130,131]. In vitro, mouse MCs can be also produced by bone marrow precursor stimulation with granulocyte-macrophage colony-stimulating factor (GM-CSF) [132]. Human MCs are produced in vitro by culturing human monocytes in the presence of GM-CSF and IL-4 [133].

### 3.2. Dendritic Cell Subpopulations

#### 3.2.1. Conventional DCs (cDCs)

##### cDC1

cDC1 is a cDC subpopulation that efficiently primes CD8+ T cells by performing antigen cross-presentation [134]. cDC1can be found both in the periphery (where they represent 30% of cDC) and in lymphoid organs (where they account for 40%).

In humans and mice, cDC1s express CD141, the chemokine receptor XCR1, C-type lectin CLEC9A, and the cell adhesion molecule CADM1 [135] (Figure 5). Moreover, in mice, cDC1s are identified by the expression of CD8α in the spleen and CD103 in non-lymphoid tissues [135] (Figure 5) and can be also characterized by the expression of other C-type lectin receptors, such as CD205 and CD207 [136].

For the generation of cDC1s, the main transcription factors (TFs) are the basic leucine zipper transcriptional factor ATF-like 3 (BATF3) [137] and IFN-regulatory factor 8 (IRF8) [138]. In mice, in addition to BATF3 and IRF8, other TFs are also essential for cDC1s generation [135,139,140], such as DNA binding protein inhibitor ID2 and nuclear factor interleukin-3-regulated protein (NFIL3).

##### cDC2

cDC2 is a more heterogeneous DC subset than cDC1 that has been shown to induce Th1, Th2, and Th17 responses from CD4+ helper T cells (Th) [141,142] and that has different regulatory roles by inducing regulatory T cells (Tregs) in some tissues, such as intestine or liver [143,144]. cDC2s can be found in lymphoid, non-lymphoid tissues, and blood [103,145] where they are more abundant than cDC1s.

This subpopulation is characterized by the expression of SIRPα in both humans and mice and CD1c or CD11b in humans or mice, respectively [111,135] (Figure 5). Moreover, cDC2s can express other markers according to their location, which produces their great heterogeneity [103].

For cDC2 differentiation, different TFs are involved, with IRF4 traditionally considered the most important [146]. However, more recent studies suggest that IRF4 is more essential for cDC2 function regulation rather than for cDC2 development [147]. Other TFs shown to be associated with cDC2 differentiation are PU.1 and RelB [148,149] in mice and IRF8 [150] in humans.

#### 3.2.2. Plasmacytoid DCs (pDCs)

pDC1 is a DC subpopulation that secretes high levels of IFN-α after TLR7/9 stimulation and has a pivotal role in viral infections [151]. In addition, these cells have also been associated with immune tolerance [152]. pDCs are continuously generated in the bone marrow and subsequently enter the bloodstream (where they constitute less than 1% of mononuclear cells) [153,154] to then home primary and secondary lymphoid tissues in steady-state [155,156,157,158]. During microbial infections or autoimmune diseases, these cells are recruited to peripheral tissues where they are typically absent [159,160]. However, although in low densities, few peripheral tissues host pDC during steady-state, such as the lung or the vagina [161,162].

Phenotypically, pDCs are distinct in mice and humans. In humans, pDCs are identified by their expression of CD123, CD303 (BDCA2), and CD304, while in mice, pDCs express CD307, B220, and SiglecH [135] (Figure 5). Importantly, human pDCs do not express CD11c [152,163] and can be divided into two subsets based on their levels of DC2 expression [164].

Regarding TFs for pDC generation, in both humans and mice, E2.2 seems to be the most important [103,111,135].

#### 3.2.3. Langerhans Cells (LCs)

LCs are a subset of cells located in epidermal surfaces, being the most numerous antigen-presenting cells in human skin [111,135]. LCs can induce several immune responses by stimulating CD4+ T cell proliferation and polarization towards the Th2 phenotype [111,165], and particularly in humans, these cells can also stimulate naïve CD8+T cells [165].

Human and murine LCs express EPCAM, low CD11c, and langerin, which act as a receptor for microbial pathogens [135] (Figure 5). Furthermore, in both humans and mice, LCs present Birbeck granules [166], a type of organelle whose function still remains unclear. Additionally, in humans, LCs are also CD1a+ and CD1c+ [135,167].

The development of LCs is mainly dependent on Runx3 and PU.1 [168,169].

#### 3.2.4. Monocyte-Derived DCs (MCs)

MCs have contributed significantly to the knowledge about DCs in humans. Due to their potential, they are currently being studied for the treatment and monitoring of different human diseases, mainly cancer [103]. Ontogeny data suggest that inflammatory DCs (another DC subpopulation that expresses high levels of CD11c and MHC-II) are the in vivo counterparts of MCs [135,170] (Figure 5).

PU.1 and IRF4 act as TFs for human monocyte differentiation into MCs in vitro [171,172].

### 3.3. Resident Dendritic Cell Distribution in the Cornea

Traditionally, the cornea has been considered a tissue devoid of resident immune cells. However, heretofore many studies in humans and mice have shown the presence of macrophages [91,93,173] and the major DC subsets (Figure 6) in this tissue [174,175]. In addition, other immune cells, such as γδ T lymphocytes [176] or NK cells [177], have been reported in the limbus.

The density of DCs decreases from the limbus towards the center of the cornea [174,175] and correlates with ocular inflammation [173]. DC localization throughout corneal layers depends on the DC subpopulation [92,94,173] (Figure 6). cDCs were traditionally thought to be confined to the peripheral cornea and the limbus. Nevertheless, it was later demonstrated that these cells are also located in the central corneal epithelium and stroma, especially during inflammation [173]. Immature cDCs are more numerous in the periphery, but in response to an inflammatory stimulus, they increase and mature throughout the cornea, with higher levels of MHC-II and co-stimulatory molecules [85,173]. LCs have been reported only in the peripheral epithelium [92] with a distribution pattern very similar to that in skin [178] and pDC are located in the anterior stroma, immediately below the basal epithelium both in the central and peripheral cornea [152].

### 3.4. Dendritic Cells in Ocular Diseases

#### 3.4.1. Humans

DCs are the major immune cells involved in the most common ocular surface diseases [179]. In humans, most of the current investigations into corneal DCs are done by using In Vivo Confocal Microscopy (IVCM), which allows the study of subbasal and stromal corneal nerves. IVCM has been more and more used for the diagnosis and management of different corneal diseases because it is minimally invasive and has a high resolution [180]. In this context, the in vivo dynamic assessment of corneal inflammatory cell density seems to be a good indicator of the disease severity. An inverse correlation between corneal nerve density and density of DCs has been described in patients with long COVID, especially those with neurological symptoms [181], and also in patients with infectious keratitis including fungal, bacterial, and *Acanthamoeba* keratitis [182] Moreover, the reduction in corneal nerves and the increase in DC density was bilateral even after unilateral infectious keratitis [183]. Along the same lines, an increase in corneal DC density has also been observed by IVCM in patients with herpetic uveitis [184], aqueous-deficient dry eye disease (DED) [185], Sjögren’s syndrome (SS) [186], and in contact lens wearers [187].

In this vein, other clinical studies have postulated a connection between the patient’s tear cytokines and corneal DCs. A significant correlation between proinflammatory cytokines and increased DC density, and reduction in corneal subbasal nerve density has been described in bacterial keratitis [188]. Significantly higher levels of IL-1β, IL-6, and IL-8 cytokines were found in tears of the affected eyes compared with healthy controls, as well as higher levels of CCL-2, IL-10, and IL-17a cytokines in the contralateral eyes [188]. However, this is not the only study that correlates proinflammatory tear cytokines with corneal DC density. In patients with rheumatoid arthritis after systemic therapy [189] a decrease in tear IL-1 and IL-6 levels is accompanied by a decrease in DC density. Table 1 summarizes these examples of ocular pathologies in which DCs are involved.

#### 3.4.2. Mice

Most of the current knowledge on the pathophysiology of DCs in ocular diseases arises from studies performed in mice. The implication of DCs in infectious keratitis and, particularly, in herpes simplex virus (HSV) keratitis, is one of the most characterized in models. As early as one day after HSV-1 inoculation, pDCs are increased in both peripheral and central cornea, and this increase progresses until 6 days post-inoculation [190]. Moreover, following HSV-1 inoculation, pDCs also increase in the draining lymph nodes (dLNs), with a major shift towards mature pDCs [194]. During primary herpes simplex keratitis, there is a neuro-invasion of sensory corneal nerves by HSV that remains latent in the TG. If there is a virus reactivation, that leads to chronic recurrent herpes stromal keratitis [195], it may eventually produce severe corneal scarring. In this context, it has been shown that pDC depletion prior to HSV-1 inoculation produces increased virus titers in the cornea and increased viral transmission to TG and dLNs [190,196], suggesting a protective role of these cells. This is the opposite of what is observed in the local depletion of cDCs, which produces a decreased corneal nerve infection and a decreased and delayed systemic viral transmission in TG and dLNs [197]. Nevertheless, in both cDC and pDC depleted-mice, a higher clinical keratitis severity was observed compared to sham-depleted animals, maybe due to the major influx of immune cells to the cornea. Further, depletion of corneal pDCs in BDCA-2-DTR mice prior to HSV-1 inoculation is accompanied by alterations in the dLN cytokine milieu, leading to decreased density of Tregs [196], as well as increased recruitment of ex-Tregs to the cornea and dLN in vivo [152].

DC immune responses in the cornea have been also widely studied in sterile models of inflammation. In ocular tissues, this sterile inflammation occurs in response to chemical and mechanical traumas, contact lens wear, or allergens. It has been shown that depletion of corneal pDCs prior to suture placement is accompanied by enhanced clinical opacity of the cornea, as well as augmented influx of inflammatory immune CD45+ cells [196,198]. Moreover, in a mouse model of ocular allergy, CD11b+ DC subset seems to play a dominant role in secondary allergic immune responses [191].

The implication of DCs in ocular diseases has also been described in diabetic sensory neuropathy in the cornea and DED. Sensory nerve density and DC populations were dramatically decreased in diabetic mice and DC decrease during wound healing results in the reduction in tissue levels of CNTF, which, in turn, impairs sensory nerve innervation and regeneration [192]. Furthermore, in an experimental model of DED induced by subcutaneous injections of scopolamine, DCs in dLNs were shown to be more activated than in control mice, suggesting that they may stimulate the T cells that participate in the onset and progression of the disease [193].

These examples (summarized in Table 1) are only a part of the huge number of results that are currently being obtained regarding the involvement of DCs in the pathophysiology of ocular diseases. However, we still have limited knowledge in many aspects of the immune response of DCs. Further studies need to be undertaken to define the molecular mechanisms behind DC immune responses and to elucidate the contribution of DCs in other ocular diseases.

## 4. Transgenic Mice to Study Neuro–Immune Interactions in the Cornea

Resident APCs in the cornea are located close to the corneal nerves [1,152,182,199,200,201]. Different studies have shown that the association between corneal nerves and APCs seems to have a potential role in corneal health and disease, with a reduced association under injury or pathological conditions [200]. In order to further analyze this neuro–immune crosstalk under pathological and steady-state conditions, our lab developed a mice model with C57BL/6J background in which sensory axons are labeled with tdTomato and DCs are labeled with green fluorescent protein (GFP) (Figure 7 and Figure 8).

To generate a mouse where the corneas have sensory axons labeled with tdTomato, we crossed Advillin-Cre^tg/+^ (Tg(Avil-cre)1Phep) mice ([202]; MGI:5292346) with Ai14^fl/fl^ (Gt(ROSA)26Sor^tm14(CAG-tdTomato)Hze^) mice ([203]; MGI:3809524) (Figure 7A). The F1 offspring was back-crossed with Ai14^fl/fl^ mice to obtain Advillin-Cre^tg/+^ Ai14^fl/fl^ offspring (F2 generation) (Figure 7B).

After generating the mice with tdTomato-labeled sensory nerves, we wanted to add GFP-labeled DC. For this purpose, Advillin-Cre^tg/+^ Ai14^fl/fl^ mice were crossed with CD11c-DTR/GFP^tg/tg^ (1700016L21Rik^Tg(Itgax-DTR/EGFP)57Lan^) mice ([204]; MGI:3057163) to obtain Advillin-Cre^tg/+^Ai14^fl/+^CD11c-DTR/GFP^tg/+^ offspring (F3 generation) (Figure 7C). CD11c-DTR mice, in addition, carry a transgene encoding for a simian diphtheria toxin (DT) receptor (DTR) plus a green fluorescent protein (GFP) fusion protein under the control of the murine CD11c promoter, which makes CD11c+ cells sensitive to DT (Figure 7D).

Ex vivo and in vivo imaging of the corneas of this mouse strain allowed a precise description of the interactions between nerve fibers and DCs, expressing tdTomato and GFP, respectively. Preliminary data on the influence of presence or absence of DCs on corneal nerve activity and nociceptive behavior of CD11c-DTR/GFP^tg/tg^ mice have been reported elsewhere [205].

## Figures and Tables

**Figure 1 ijms-23-02997-f001:**
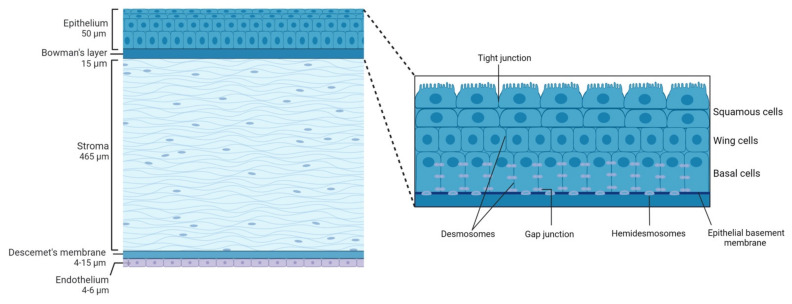
Schematic representation of the corneal structure, showing its five main layers and their thickness. Inset: diagram of the corneal epithelium, showing its different cell layers and details on the different types of cell-cell junctions contributing to corneal epithelium impermeability.

**Figure 3 ijms-23-02997-f003:**
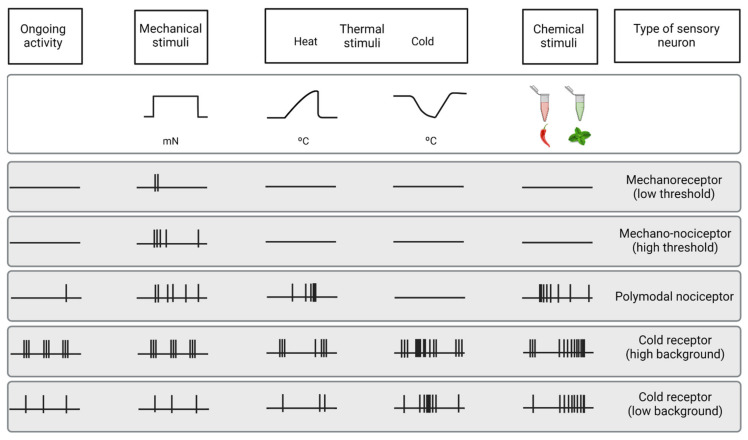
Functional types of sensory neurons innervating the cornea. Schematic representation of the spontaneous and stimulus-evoked nerve impulse activity of the different functional types of sensory nerves innervating the cornea. Based on the characteristics of the impulse discharge in absence of intended stimulation (ongoing activity) and response to different types of stimuli (upper part of the figure), the peripheral terminals of primary sensory neurons innervating the cornea are classified into five different functional types of sensory neurons.

**Figure 4 ijms-23-02997-f004:**
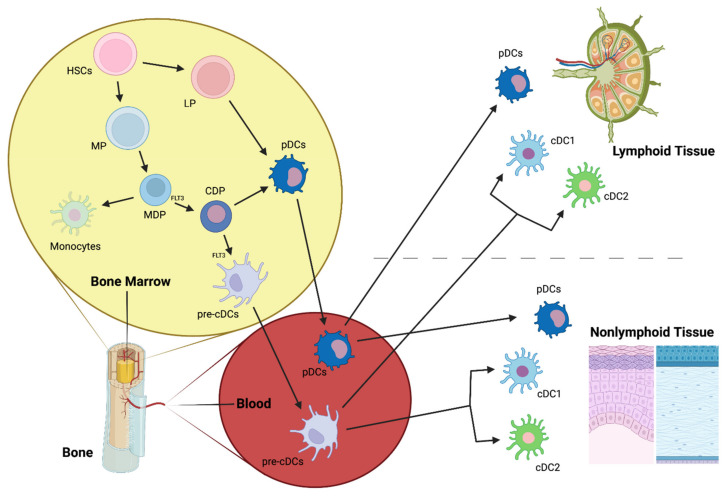
cDC1 and cDC2, main conventional dendritic cell subtypes; CDP, common DC precursor; FLT3, Fms-Related Tyrosine Kinase 3; HSCs, hematopoietic stem cells; LP, lymphoid precursor; MDP, macrophage-DC precursor; MP, myeloid precursor; pre-cDCs, pre-classical dendritic cells; pDCs, plasmacytoid dendritic cells.

**Figure 5 ijms-23-02997-f005:**
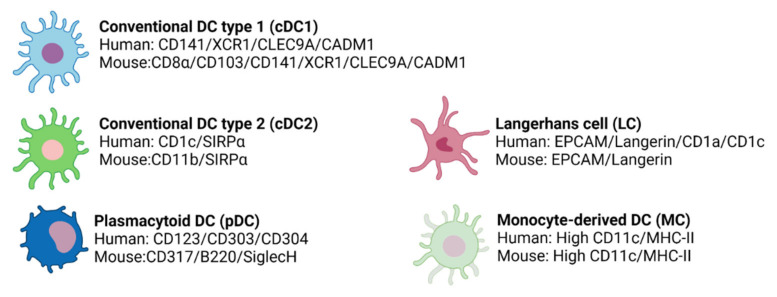
The different DC subtypes express specific surface markers in humans and mice.

**Figure 6 ijms-23-02997-f006:**
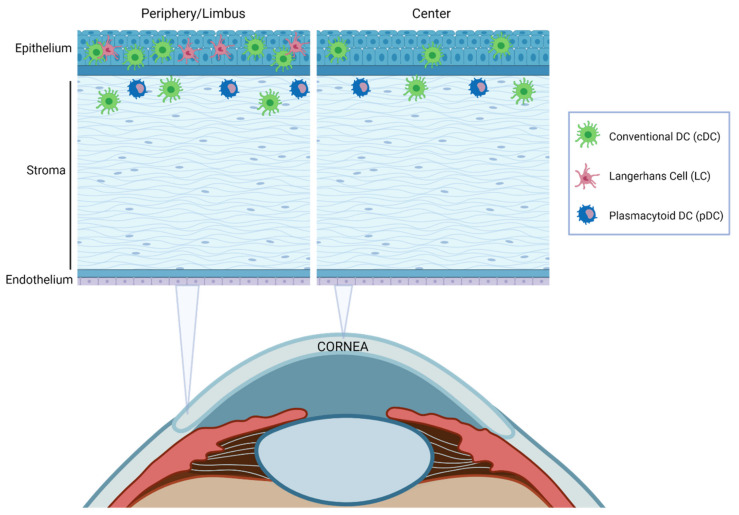
Schematic illustration of the distribution of resident DCs in the cornea at steady-state. Most DC subtypes are preferentially distributed in the epithelium and the anterior stroma and are more abundant in the peripheral cornea and limbal area than in the central cornea.

**Figure 7 ijms-23-02997-f007:**
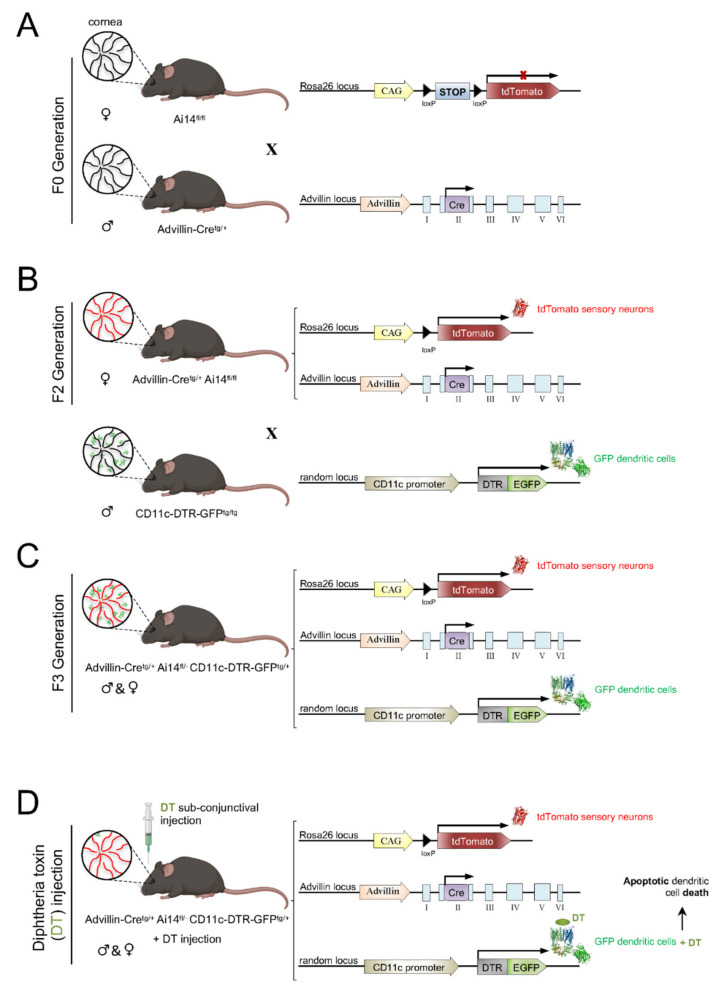
Scheme of the generation of the Advillin-Cretg/+ Ai14fl/- CD11c-DTR-GFPtg/+ mouse model. Ai14fl/fl females were crossed with Avidillin-Cretg/+ males to produce the F1 generation (**A**). To obtain a F2 generation of Advillin-Cretg/+ Ai14fl/fl mice (**B**), F1 females were crossed with Ai14fl/fl males (which had corneal sensory nerves framed with tdTomato). To generate the animal model of interest (**C**), F2 females were crossed with male CD11c-DTR-GFPtg/tg mice that expressed Diphtheria toxin Receptor (DTR) and GFP in DCs. Subconjunctival administration of DT to the resultant Advillin-Cretg/+ Ai14fl/− CD11c-DTR-GFPtg/+ mice caused apoptosis of corneal DCs (**D**).

**Figure 8 ijms-23-02997-f008:**
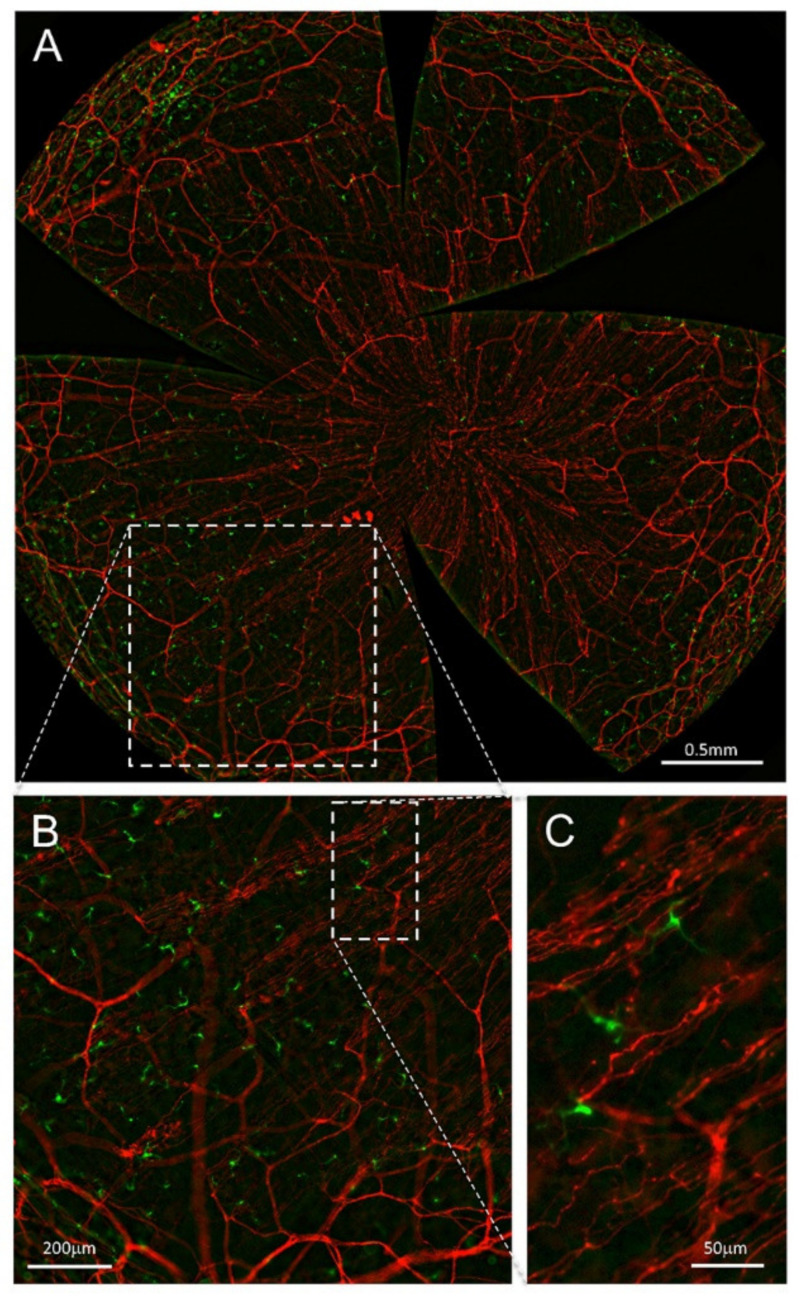
Nerves and DCs in the advillin-Cretg/+ Ai14fl/- CD11c-DTR-GFPtg/+ mouse cornea. All panels are maximal intensity z-projection stacks (30µm) performed with a Leica THUNDER Imager Tissue provided with a 20× objective. (**A**) Flat mount overview of a whole-mount cornea (64 tiles merged) showing tdTomato (red) expression in corneal sensory nerves and GFP (green) expression in DCs after computational clearing of the out-of-focus blur. (**B**,**C**) Higher magnification images showing nerve fibers and DCs endogenously expressing tdTomato and GFP, respectively. Scale bars: A, 500 µm; B, 200 µm; C, 50 µm. Methods: Ocular globes from 5-month-old mutant mice were fixed for 2 h at RT in 4% paraformaldehyde in PBS and then washed in PBS. After washing, corneas were dissected and flat-mounted with Fluoromount-G.

**Table 1 ijms-23-02997-t001:** Changes in Dendritic Cells (DCs) in ocular pathologies.

Disease	Species	Corneal Changes	Reference
Long COVID	Human	Reduced corneal nerve density and increased DC density	[181]
Infectious keratitis (fungal, bacterial, or *Acanthamoeba*)	Human	Reduced corneal nerve density and increased DC density	[182]
Herpetic uveitis	Human	Increased DC density	[184]
Aqueous-deficient dry eye disease (DED)	Human	Increased DC density	[185]
Sjögren’s syndrome (SS)	Human	Increased DC density	[186]
Contact lens wearing	Human	Increased DC density	[187]
Rheumatoid arthritis	Human	Decreased DC density	[189]
Herpes simplex virus (HSV) keratitis	Mouse	Increased DC density	[190]
Ocular allergy	Mouse	DC lead to allergic T cell responses	[191]
Diabetic sensory neuropathy	Mouse	Reduced corneal nerve and DC density	[192]
Dry eye disease (DED)	Mouse	More activated DCs in lymph nodes	[193]

## Data Availability

Not applicable.

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
