# Peer review of "An Experimental Model of Neuro–Immune Interactions in the Eye: Corneal Sensory Nerves and Resident Dendritic Cells"

_ijms, 2022, doi:10.3390/ijms23062997_

Round 1
Reviewer 1 Report
Line 121: What is the role of Bowman membrane? Why will a superficial corneal injury heal with scar if Bowman membrane is involved?
Line 162: Where can be Dua layer localized?
Line 310, 682: What is the source of Fig2/Fig8? If they are your own figures, please detail the method. If they are not your own ones, you need permission.
Line 577 Please summarize the clinical relevances of dendritic cells in a table
Author Response
Thank you for your careful review of the manuscript.
Below we transcribe your questions and/or suggestions followed by our answers.
Line 121: What is the role of Bowman membrane?
Following your suggestion, we have included a new paragraph in the text (lines 128-133) to comment on this point:
“Different roles have been ascribed to this layer of the cornea that is not present in all species, such as that it provides some kind of barrier function against the passage of macromolecules such as medium and large size proteins, or that it is responsible for a substantial portion of the biomechanical rigidity of the cornea. However, there are also studies that conclude the opposite and the exact function of this layer remains unclear (see Wilson 2020 for a review (Wilson, 2020)).”
Line 121: Why will a superficial corneal injury heal with scar if Bowman membrane is involved?
Bowman’s layer is highly resistant to damage but if it becomes injured it cannot regenerate. Therefore, if Bowman's layer is involved, even if it is a superficial corneal injury, scarring is more than likely to occur. We have included a new paragraph (lines 133-138) to state this point:
“It has been hypothesized that Bowman’s layer develops because of cytokine-mediated interactions occurring between corneal epithelial cells and the underlying keratocytes, including negative chemotactic and apoptotic effects on the keratocytes by low levels of cytokines such as interleukin-1α (Wilson and Hong, 2000). Bowman’s layer is highly resistant to damage but it cannot regenerate after injury and may result in a scar (Lee, 2016).”
Line 162: Where can be Dua layer localized?
Thank you for pointing out that our review did not include a specific comment about pre-Descemet or Dua’s layer. We have now included a paragraph in the text, section 2.1.3 (lines 163-166):
“In the posterior part of the human corneal stroma, there is a distinct layer that constitutes the separation between the stroma and the Descemet’s membrane. This pre-Descemet or Dua’s layer is acellular and composed of 5 to 8 lamellae of predominantly type-1 collagen bundles arranged in transverse, longitudinal, and oblique directions (Dua et al., 2013). “
Line 310, 682: What is the source of Fig2/Fig8? If they are your own figures, please detail the method. If they are not your own ones, you need permission
Thank you for your comment, this will improve our draft. All figures included in this review are our own figures, including figures 2 and 8. To make it clearer, we have specified it in the figure legends. We have also added a short protocol of how the tissue has been processed in each figure. We added:
In Figure 2 legend:
Methods: C57BL/6J eyes were fixed 2 h at RT in methanol and DMSO (4:1), incubated 5 min at -20°C in methanol, rehydrated and washed in PBS. (A-F) Corneas were dissected, incubated 2 h at RT in blocking solution (5 % goat normal serum, 1% BSA and 0.1%Triton X-100 in PBS) and 48 h at 4°C with anti-ꞵ tubulin III antibody (1:500 in blocking solution; #801201, BioLegend), rinsed and then incubated for 2 h at RT with AF555 (1:500 in PBS; #A32727, Thermo Fisher Scientific), washed in PBS and mounted with Fluoromount-G (Thermo Fisher Scientific). (G-I) Eyes were cryoprotected (30% sucrose in PBS overnight at 4°C), embedded in OCT, frozen in liquid nitrogen, cut on a cryostat in serial 15 mm thick sections, and mounted in slides. Tissue sections were rinsed in 0.03% Triton X-100 in PBS and then for 30 min in blocking solution followed by overnight incubation at 4°C with anti-ꞵ tubulin III antibody in blocking solution, washing with PBS and incubation with AF555 in PBS for 2 h at RT. Afterwards, slides were rinsed with PBS, incubated with Hoechst 33342 (10 mm/ml; Molecular Probes), and coverslipped with Fluoromount-G. Images were collected using a laser scanning confocal microscope Zeiss LSM 880.
In Figure 8 legend:
Methods: Ocular globes from 5 months old mutant mice were fixed 2 h at RT in 4% paraformaldehyde in PBS and then washed in PBS. After washing, corneas were dissected and flat mounted with Fluoromount-G.
Line 577: Please summarize the clinical relevances of dendritic cells in a table
Thank you very much for this suggestion. We have included Table 1 summarizing the clinical relevance of dendritic cells with the examples of ocular diseases that we have described in the review for both humans and mice.
Reviewer 2 Report
This review entitle An Experimental Model of Neuro-immune Interactions in the 2
Eye: Corneal Sensory Nerves and Resident Dendritic Cells by Laura Frutos-Rincón et al is written very well also organized and described superbly. The figures in the review also clearly express the subject. I feel that this work is scientifically sound and should be accept in present form.
Author Response
Thank you for reviewing the manuscript and for your kind words about our review. We are very pleased that it was to your liking. Your assessment gives much more importance to the work and serves as a motivation for us to continue our work in the lab, in particular to the PhD students involved in this project.
English has been revised throughout the manuscript and several spelling mistakes have been fixed.